# Embodied CO$_2$ Emission Changes in Manufacturing Trade: Structural Decomposition Analysis of China, Japan, and Korea

**Young Yoon [1], Yoon-Kyung Kim [2] and Jinsoo Kim [1,\*]**

1. Department of Earth Resources and Environmental Engineering, Hanyang University, Wangsimni-ro 222, Seoul 04763, Korea; youngyoon@hanyang.ac.kr
2. Department of Economics, Ewha Womans University, Ewhayeodae-gil 52, Seoul 03760, Korea; yoonkkim@ewha.ac.kr
* Correspondence: jinsookim@hanayng.ac.kr; Tel.: +82-2-2220-2241; Fax: +82-2-2220-4769

**Abstract:** This study investigated the driving factors of embodied carbon emission changes in manufacturing trades through structural decomposition analysis. For empirical analysis, we developed an environmental multiregional input–output model for Korea, Japan, and China for 1995–2009. The three countries, which are economically and environmentally significant in Asia, are not only tightly linked economically through global value chains, but also close geographically, sharing various environmental issues. The results show that China is a net exporter of embodied carbon emissions to Japan and Korea, despite a substantial trade deficit. Its exports are more carbon-intensive than its imports from Japan and Korea. China's embodied emissions were mainly affected by a change in carbon-intensive production and trade structure, and Japan's and Korea's were affected by China's final demand. At the sectoral level, "Electrical and Optical Equipment", "Basic Metals and Fabricated Metal", and "Textiles and Textile Products" mainly affected the embodied carbon emission changes in these three countries. As a result, a considerable share of carbon-intensive production has shifted to China and increased consumption of China's final products and services in the manufacturing industries, resulting in a significant increase in embodied carbon emissions. Additionally, our findings at the sectoral level could provide important evidence regarding the effective environmental policies that enable sustainable industries. With the increasing interest in the embodied carbon emissions, future research would pay more attention to the bilateral trades of major carbon-emitting countries and multilateral trades.

**Keywords:** embodied carbon emissions; manufacturing; structural decomposition analysis; bilateral trade; environmental multiregional input–output; carbon policies; emission reduction

## 1. Introduction

Over the last two decades, the volume of international trade has increased greatly due to globalization and trade liberalization as initiated by the World Trade Organization (WTO). Due to the expansion of vertical specialization of production, the interdependence of the global economy has greatly increased, and much of the goods and services consumed by producers and consumers in one country are imported rather than produced domestically. However, this expansion of trade exerts adverse environmental impacts by, for example, transferring the source of environmental pollution from domestic to overseas areas. In particular, under the current Kyoto Protocol, territorial-based national greenhouse gas (GHG) inventories assign responsibility for GHG emissions to the producing countries, and countries with reduction obligations have been provided incentives to reduce their emissions through international trade with other countries with less stringent emission constraints [1].

In the Paris Agreement, which is a bottom-up approach to implementing countries' voluntary emission goals and reduction targets, balanced environmental regulations cannot be expected between countries. In addition, because of the different economic interests of developed and developing countries, the possibility of carbon leakage is expected to be high in the new climate system. Such carbon leakage not only reduces the effectiveness of international mitigation policies, but also has a negative impact on industrial competitiveness and raises the issue of inequity in emission responsibility between developed and developing countries. Therefore, it is necessary to consider the global supply chain and carbon emission in terms of consumption when designing an optimal global environmental policy for GHG reduction.

Asia is one of the world's fastest-growing regions regarding carbon emissions. Among Asian countries, China, Japan, and Korea are the first (28%), fifth (3.4%), and seventh highest emitters (2.1%) of $CO_2$, respectively, making up a large share of the global $CO_2$ emissions [2]. The three countries, which are economically and environmentally significant in Asia, are not only tightly linked economically, but also close geographically, sharing various environmental issues. Since the 1990s, these three countries have continued expansion of trade and vertical division of labor for manufacturing. Given that increasing manufacturing trade among these three countries and the resulting production and consumption are major sources of carbon emissions, effective carbon reduction needs to consider embodied carbon emission in trade and key driving factors of changes in this region.

Considering the increasing influence of international trade on the global environment, analysis of embodied emissions in trade has become an important issue in addressing climate change [3]. Many studies have estimated carbon emissions embodied in international trade based on the input–output model. They have shown growing embodied emissions in trade and an increasing influence of international trade on national emission trends. For example, Peters, et al. [4] found that the embodied emissions in traded goods and services increased from 4.3 Gt $CO_2$ in 1990 (20% of global emissions) to 7.8 Gt $CO_2$ in 2008 (26%). The net emissions transfer via international trade from developing to developed countries increased 17% per year in average growth, from 0.4 Gt $CO_2$ in 1990 to 1.6 Gt $CO_2$ in 2008, which exceeds the Kyoto Protocol emission reduction target. Xu and Dietzenbacher [5] found that the total emissions from production increased by 32% from 19.0 Gt in 1995 to 25.3 Gt in 2007, while the emissions embodied in traded goods and services increased by 80% from 4.6 Gt in 1995 to 8.3 Gt in 2006. This indicated that international trade is a significant factor for the change in emissions.

Given the importance of carbon emissions in Asia in global emission levels, many studies have focused on issues associated with trade within Asian countries. Zhao, et al. [6] investigated the $CO_2$ emissions embodied in trade between China and Japan by using the input–output approach. The authors showed a significant increase in $CO_2$ emissions embodied in the China–Japan trade, and $CO_2$ emissions embodied in China's exports increased by about 100% from 1995 to 2009. Du, et al. [7] analyzed the embodied $CO_2$ emissions in the China–US trade and revealed that the increased embodied emissions could mostly be attributed to the increase in trade volume over the past decade.

To more specifically analyze the carbon flow embodied in international trade, some studies examined the driving factors for the change in the emissions using structural decomposition analysis (SDA), which assesses direct and indirect emissions and applies the environmental input–output method. Su, et al. [8] and Su and Ang [9] analyzed the driving factors of changes for $CO_2$ emissions embodied in China's foreign trade. Du, et al. [7] and Yang, et al. [10] examined the changes in embodied carbon emissions in trade among China and its major trade partners by using the Embodied Emissions in Bilateral Trade (EEBT) model. Xu and Dietzenbacher [5] determined the decomposition of changes for $CO_2$ emissions embodied in the foreign trades of 40 countries. However, given the easier application and transparency of EEBT, most studies have conducted decomposition analyses with the EEBT model in bilateral trade [7]. Without considering features such as increased intermediate goods trade through vertical specialization, their focus was mainly on estimating the impact on total exports using the

domestic technical assumption. They did not consider the emissions embodied through interregional feedback effects and the expanded global supply chain.

In this study, we determined the carbon emissions embodied in manufacturing trade among China, Japan, and Korea through a Multiregional Input–Output (MRIO) approach. As one of the most energy-consuming industries, the manufacturing sector is contributing significantly to the increase in GHG emissions in most countries [11]. Using this model, not only can the analysis be more detailed than the EEBT approach, but the international feedback effect through the global supply chain in the manufacturing sector can also be reflected. We also used the SDA technique to analyze the driving factors of changes in embodied emissions during 1995–2009 with the World Input–Output Database (WIOD). The WIOD is the most common and widely used input–output database for the relationship for inter-industries and the assessment of embodied flows [12]. The advantage of the WIOD is that it provides integrated environmental variables and enables the capture of yearly changes in environmental factors.

By examining the effects of trade on carbon emissions in each country through the estimation of the emission balance, we identified the carbon leakage and derived important implications for climate policy. Given the economic size, $CO_2$ emission intensity, trade expansion, and position of the three countries in the Kyoto Protocol, understanding the flow of trade in this region and the carbon emissions in their bilateral trade is an important step in international climate change discussions. These findings can help reduce GHG in not only these three countries, but also in Northeast Asia, and ultimately help establish global carbon reduction policies.

The rest of this paper is organized as follows: Sections 2 and 3 present the applied empirical methods and data characteristics. Section 4 describes the empirical results of $CO_2$ emissions embodied in manufacturing trade among the three countries and the driving factors for the changes in embodied emissions. Finally, Section 5 summarizes the discussion and conclusions for the results and provides potential policy implications.

## 2. Methodology

To analyze the carbon emissions in trade and the driving factors of them, we used input–output analysis. The input–output analysis was originated by Leontief [13]. The input–output model is widely applied in embodied carbon flow studies up to this day because of its advantage of considering the direct and indirect emissions induced by industrial linkages [14]. Additionally, we used SDA to analyze the driving factors of carbon emissions in trade based on the input–output model. Therefore, we first constructed an environmental input–output model and then decomposed the equations, the subsection below deals with the detailed environmental input–output model and the decomposition process of the driving factors.

### 2.1. Environmental Multiregional Input–Output Model

Global economic integration through international trade has increased the complexity of national GHG reduction policies. MRIO is a useful tool for quantifying the effects of different countries' production and consumption in complex trade relations. Expanding the single input–output matrix, the MRIO treats the global economy as a single economy [15]. In this study, extending the model developed in Serrano and Dietzenbacher [16], the total carbon emissions embodied in exports (ECE) from country $c$ at the sectoral level is as follows:

$$ECE^c = \left[\sum_{k=1}^{N}\left(P^{kc}\right)\right]\left(\sum_{j\neq c}^{N}Y^{cj}\right) + \sum_{j\neq c}^{N}\left[\left(K^{cj}\right)\left(\sum_{k=1}^{N}Y^{jk}\right)\right] \tag{1}$$

where $P^{kc}$ is a row vector, denoting the direct and indirect emissions generated in a sector in one country per unit of final goods and services produced in a specific sector in country $c$, $K^{cj}$ is a matrix, denoting the direct and indirect emissions generated in a specific sector in one country per unit of final goods and services produced in a sector in country $j$, and $Y$ represents the final demand.

Carbon emissions in exports can be divided into two parts. One (left) is carbon emissions induced by the exports of final products, which include the emissions generated in the world. The other (right) is carbon emissions induced by exports of intermediate products, which include the emissions generated in country $c$. Therefore, the carbon emissions embodied in exports include domestic carbon emissions in country $c$ and foreign carbon emissions in country $k$. Similarly, emissions embodied in imports (ECI) can be calculated as follows:

$$ECI^c = \sum_{j \neq c}^{N} \left[ \sum_{k=1}^{N} \left( P^{kj} \right) \right] \left( Y^{jc} \right) + \left[ \sum_{j \neq c}^{N} \left( K^{jc} \right) \right] \left( \sum_{k=1}^{N} Y^{ck} \right) \tag{2}$$

Next, we calculated the carbon emission balance (EB) by subtracting imports from exports, net $CO_2$ emissions from $c$ to $j$, as follows:

$$
\begin{aligned}
EB^{cj} &= \left[ \sum_{k \neq j}^{N} \left( P^{kc} \right) \right] \left( \sum_{j \neq c}^{N} Y^{cj} \right) + \sum_{j \neq c}^{N} \left[ \left( K^{cj} \right) \left( \sum_{k \neq c}^{N} Y^{jk} \right) \right] \\
&- \sum_{j \neq c}^{N} \left[ \sum_{k \neq c}^{N} \left( P^{kj} \right) \right] \left( Y^{jc} \right) - \left[ \sum_{j \neq c}^{N} \left( K^{jc} \right) \right] \left( \sum_{k \neq j}^{N} Y^{ck} \right)
\end{aligned}
\tag{3}
$$

In the above emission balance equation, foreign emissions in the import of intermediate products re-exported as final products for country $c$ and domestic emissions in the export of intermediate products reimported as final products for country $j$, existing in both exports- and imports-equation, are eliminated. It reflects the emissions that are first imported as intermediate products and then exported as final products again. To analyze the bilateral effect, we assumed an economy consisting of two countries. When considering only two economies, the interpretation of the analysis of the changes in carbon emissions embodied in trade is clear; this is covered in more detail in the following sections. To obtain the yearly changes in ECE, ECI without the price effect, we subtracted only the volume of embodied emissions in year $t$ in current prices from that in year $t+1$ in the previous year's prices and added the volume to year $t$.

## 2.2. Structural Decomposition Analysis

To decompose the driving factors of embodied emission changes over time, we used SDA, an input–output table-based decomposition methodology. The formula for calculating the carbon emissions embodied in international trade consists of two factors: embodied emissions in trade for one unit of final goods $P$, and exports $Y$. First, $P$ can be divided into a $CO_2$ intensity matrix $C$ and Leontief inverse matrix $L$. First, $C$ can be further divided as follows:

$$C = \left[ u \left( Q^c \otimes W^c \right) \otimes E^c \right]_{1 \times N} \tag{4}$$

where the $k \times m$ matrix $Q$ includes energy emission factors, with the element $q_{ei}$ representing the carbon emissions per unit heat of energy $e$ in sector $i$; the $k \times m$ matrix $W$ representing the energy consumption structure, with element $w_{ei}$ representing the share of energy $e$ consumption in the total energy consumption of sector $i$; the $m \times 1$ vector $E$ denotes energy intensities, with element $e_i$ representing the total energy consumption per unit output of sector $i$; $u$ is a $k \times 1$ vector of 1s; and $\otimes$ is the Hadamard product of the matrices.

Matrix $A$ reflects the input coefficients that measure the intermediate inputs per unit of output. Regardless of where they are sourced from, the total intermediate inputs required for a unit of output eventually refers to production technology. If the total intermediate inputs are $H$, then matrix $A$ can be further decomposed as follows:

$$A = \left( T^{jc} \otimes H^c \right)_{N \times N} \tag{5}$$

where $H^c = \sum_{j=1}^{N} A^{jc}$ represents the production technology of country $c$ with the $m \times m$ matrix and element $h_{ij}^c$ denoting the total intermediate inputs of sector $j$ for sector $i$ per unit output; $T^{jc}$ represents the composition of intermediate goods and services with the $m \times m$ matrix and element $t_{ij}^{jc} = a_{ij}^{jc} / h_{ij}^c$

denoting the share of intermediates of sector $i$ in country $j$ for total intermediates $H$ of sector $j$ in country $c$.

The final demand can also be divided into factors reflecting the total final demand and factors reflecting the final demand structure as follows:

$$Y = \left(P^{ck} \otimes G^k\right)_{N \times N} \tag{6}$$

where the $1 \times m$ matrix $G^k = \sum_{c=1}^{N} Y^{ck}$ represents the overall level of final demand of country $k$, with element $g_i^k$ denoting the total final demand of sector $i$ in country $k$; $P^{ck}$ represents the trade structure of final products, with the $1 \times m$ matrix and element $p_i^{ck} = y_i^{ck} / g_i^k$ reflecting the share of final products of sector $i$ in country $c$ for total final demand of sector $i$ in country $k$. Based on the above decomposition results, we can re-express the ECE as follows:

$$
\begin{aligned}
ECE^{cj} &= \left[\sum_{j=1}^{N} u(Q^j \otimes W^j) \otimes E^j \left(I - (T^{jc} \otimes H^c)\right)^{-1}\right]\left(P^{cj} \otimes G^j\right) \\
&+ \left[u(Q^c \otimes W^c) \otimes E^c \left(I - (T^{cj} \otimes H^j)\right)^{-1}\right]\left(\sum_{k=1}^{N} P^{jk} \otimes G^k\right)
\end{aligned}
\tag{7}
$$

In this study, the carbon emissions embodied in exports and imports include connected foreign and domestic production and consumption that is not included in the domestic technical assumption in some studies; thus, we can separate the factors into domestic and foreign parts. Therefore, the emissions embodied in exports depend on domestic and foreign factors. In particular, since this study focuses on bilateral trade, it is possible to interpret the driving factors of change more clearly by separating domestic and foreign factors. Equations (8) and (9) represent the theoretical framework of embodied carbon emissions in exports and imports using the aforementioned separate factors.

$$ECE = f(Q^{(c)}, Q^{(-c)}, W^{(c)}, W^{(-c)}, E^{(c)}, E^{(-c)}, T^{(c)}, T^{(-c)}, H^{(c)}, H^{(-c)}, P^{(c)}, P^{(-c)}, G^{(c)}, G^{(-c)}) \tag{8}$$

$$ECI = f(Q^{(c)}, Q^{(-c)}, W^{(c)}, W^{(-c)}, E^{(c)}, E^{(-c)}, T^{(c)}, T^{(-c)}, H^{(c)}, H^{(-c)}, P^{(c)}, P^{(-c)}, G^{(c)}, G^{(-c)}) \tag{9}$$

where the superscript $(c)$ represents the factors at home and the superscript $(-c)$ represents the factors abroad. Finally, based on the above framework, we decomposed ECE and ECI changes into the following components: changes in emission factors ($\Delta Q$), changes in energy consumption structure ($\Delta W$), changes in energy intensities ($\Delta E$), changes in the composition of intermediate goods and services ($\Delta T$), changes in the production technology ($\Delta H$), changes in the trade structure of final goods and services ($\Delta P$), and changes in the levels of final demand ($\Delta G$). Each driving factor has two parts, which distinguish changes at home and changes abroad. To solve the nonuniqueness problem in SDA, Dietzenbacher and Los [17] proposed the ideal decomposition method. However, it is difficult to conduct a calculation when the number of driving factors are many. In this study, the arithmetic average of two polar decomposition methods was taken to overcome this problem. This approximate method offers the advantage of relatively simple operation and provides a good approximation for the results of the D&L method [18]. The full equations of ECE$^{cj}$ derived from the two polar decomposition methods are given by Yoon [19].

## 3. Database for a Decomposition Analysis

To construct an environmental input–output model, information on the manufacturing production structure, international trade flows, energy consumption, and $CO_2$ emissions is needed. Several MRIO databases such as Global Trade Analysis Project (GTAP), Eora, Exiobase, and OECD Inter-Country Input–Output Tables provide information for constructing environmental input–output models. Due to differences in data sources and data integration methods, there is a possibility that the results of calculations may differ when using different databases. Arto, et al. [20] address this problem through the practical calculation results for the global carbon footprint.

In this study, the WIOD database was used to estimate the carbon emissions embodied in manufacturing trade. Compared to other databases, the WIOD not only provides the time-series multiregional input–output tables, but also the environmental satellite accounts such as energy consumption and $CO_2$ emissions at the industry level.

Therefore, we conducted the decomposition analysis on a yearly basis and captured fluctuations in driving factors in more detail than previous studies. It covers 40 countries and 35 sectors from 1995 to 2009 in the current and previous year's price tables [21]. Since the current and previous year's prices are provided simultaneously, the price effect can be controlled in the decomposition analysis. However, the database only provides information on carbon dioxide emissions from fossil fuel and energy consumption of each country until 2009, thus our analysis period is from 1995 to 2009. We could expand the environmental accounts into 2014 if we combine the other database with WIOD, but then we lose the consistency of data estimation. Additionally, the sector classifications for manufacturing industries are different. Since it is challenging to find out the analysis of embodied carbon emissions in trades correctly for 1995–2009, we chose the consistency of data rather than the expansion of the study period.

All data in the WIOD are based on the official national statistics of each country, and the input–output table for the previous year's prices was constructed using row-wise deflation and industry output deflators [5]. We applied the 14 manufacturing industrial classification provided by the WIOD; Food, Beverages, and Tobacco; Textiles and Textile Products; Leather, Leather and Footwear; Wood, Cork, and Wood Products; Pulp, Paper, Paper Printing, and Publishing; Coke, Refined Petroleum, and Nuclear Fuel; Chemicals and Manufacture of Chemical Products; Rubber and Plastics; Other Non-Metallic Minerals; Basic Metals and Fabricated Metal; Machinery, n.e.c.; Electrical and Optical Equipment; Transport Equipment; n.e.c. and Recycling. A detailed description of the WIOD database is given by Tukker and Dietzenbacher [22].

## 4. Results

### 4.1. Embodied Carbon Emissions in Manufacturing Trade among China, Japan, and Korea

$CO_2$ emissions embodied in manufacturing trade are quantified using the MRIO model. The main results of the embodied emissions in the bilateral trade among China, Japan, and Korea we obtained are as follows. First, Figure 1 shows the exports and imports in 1995 prices and carbon emissions embodied in trade between China and Japan during 1995–2009. The total $CO_2$ emissions embodied in the exports from China to Japan were 94 Mt in 1995, which decreased to 81 Mt in 2000 and then increased to 144 Mt in 2009. Conversely, $CO_2$ emissions embodied in the exports from Japan to China increased from 7 Mt in 1995 to 11 Mt in 2000 and reached 22 Mt in 2009. The most striking feature between China and Japan is the huge imbalance in embodied emissions in trade. This gap generally continues to grow, except for 1999–2000, the period affected by the Asian financial crisis, and 2007–2008, the period affected by the global financial crisis. During the period 2000–2005, imbalances increased significantly as China's carbon exports increased dramatically. Overall, China exports significant amounts of embodied carbon to Japan, while importing a relatively low amount. On the other hand, Japan's exports were higher than China's exports in all periods. As a result, China exported relatively carbon-intensive products compared to Japan. Considering the growing imbalances and that Japan's exports to China are more than China's exports to Japan, these results indicate that Japan imports many carbon-intensive products from China, and the possibility of carbon leakage was high during 2000–2005.

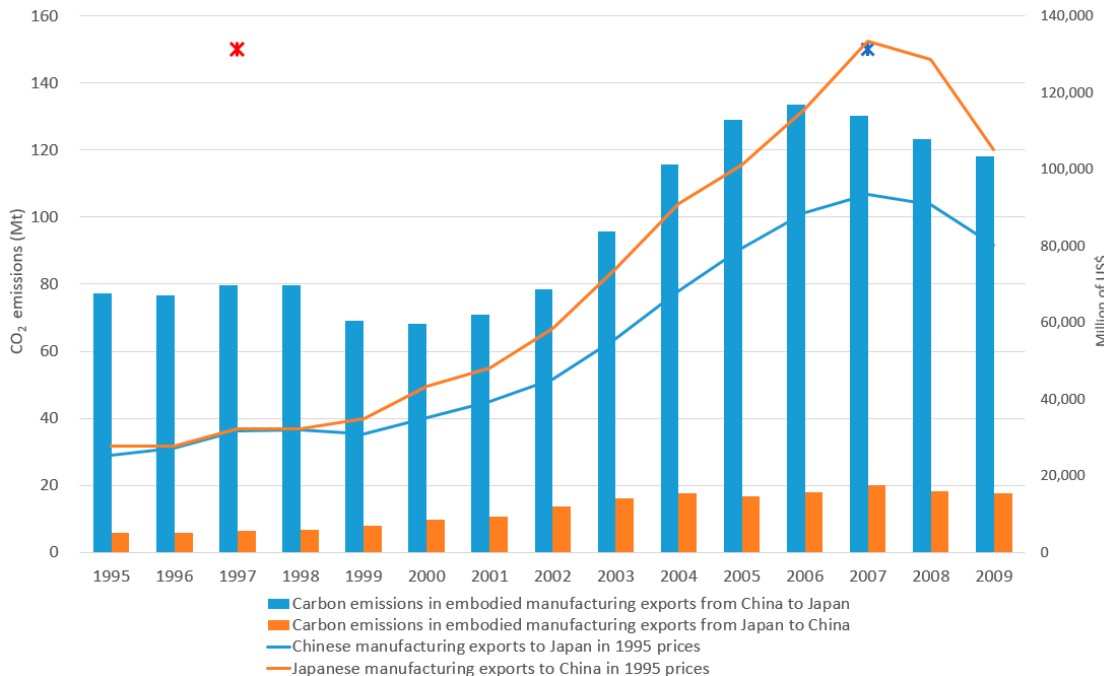

**Figure 1.** Exports and embodied CO$_2$ emissions in manufacturing trade between China and Japan. Note: A red marker indicates the Asian financial crisis, and a blue marker indicates the global financial crisis.

Second, from Figure 2, the CO$_2$ emissions embodied in the manufacturing exports from China to Korea were 17 Mt in 1995, which increased to 19 Mt in 2000 and then increased to 47 Mt in 2009. From Korea to China, CO$_2$ emissions embodied in the exports increased from 7 Mt in 1995 to 17 Mt in 2000 and reached 35 Mt in 2009. The carbon emissions embodied in exports have increased in both countries, similar to the trend for imports. However, the increases in carbon emissions embodied in Chinese exports to Korea were steeper changes compared with those embodied in Korean exports to China. The imbalance continues to grow except for 1998 and 2009, the period affected by the Asian financial crisis and the global financial crisis. During the period 2001–2008, imbalances increased significantly as China's carbon exports increased dramatically. In terms of exports and imports, Korea has a substantial trade surplus from China. These results indicate that Chinese exports are much more carbon-intensive than its imports. Considering the growing imbalances, that Korea imports many carbon-intensive goods and services from China, and various circumstances such as China's entry into the WTO and their adoption of the Kyoto Protocol, the possibility of carbon leakage was high after 2001.

Third, from Figure 3, the CO$_2$ emissions embodied in exports from Korea to Japan increased from 20 Mt in 1995 to 22 Mt in 2000 and then decreased to 13 Mt in 2009. Conversely, the CO$_2$ emissions embodied in the exports from Japan to Korea were 7 Mt in 1995, which decreased to 6 Mt in 2000 and then increased to 8 Mt in 2009. Korea has a relatively larger amount of carbon emissions embodied in manufacturing exports than Japan. However, the carbon emissions embodied in Korea's exports are decreasing, while those in Japan's exports are slightly increasing; overall, the imbalance is decreasing. However, Korea exports more compared to Japan. This means Japan is importing carbon-intensive products from Korea.

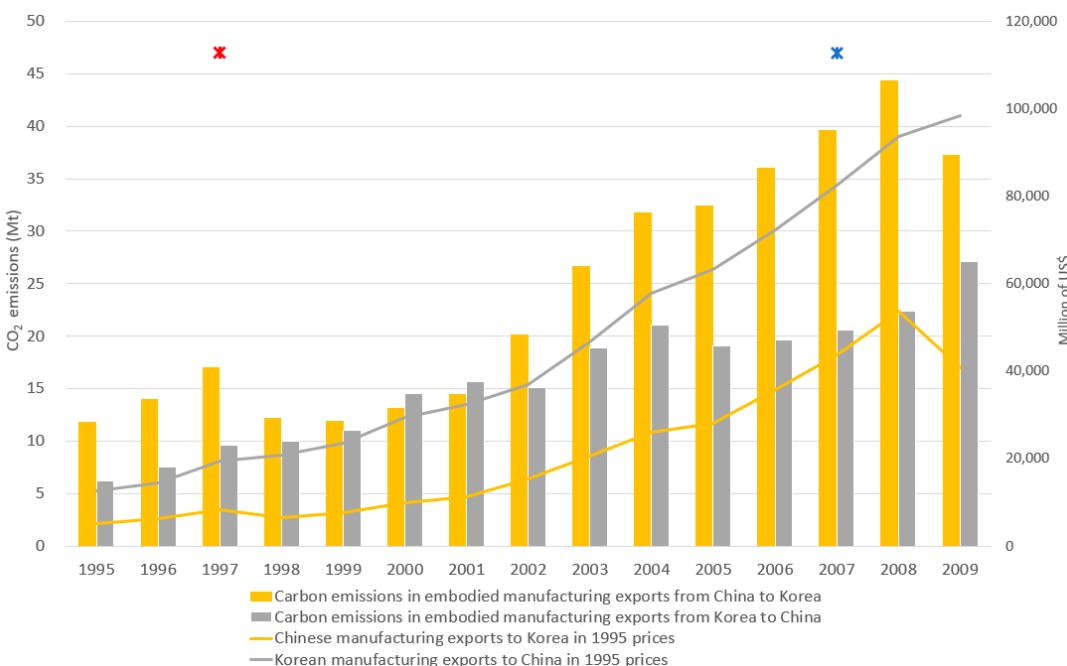

**Figure 2.** Exports and embodied $CO_2$ emissions in manufacturing trade between China and Korea. Note: A red marker indicates the Asian financial crisis, and a blue marker indicates the global financial crisis.

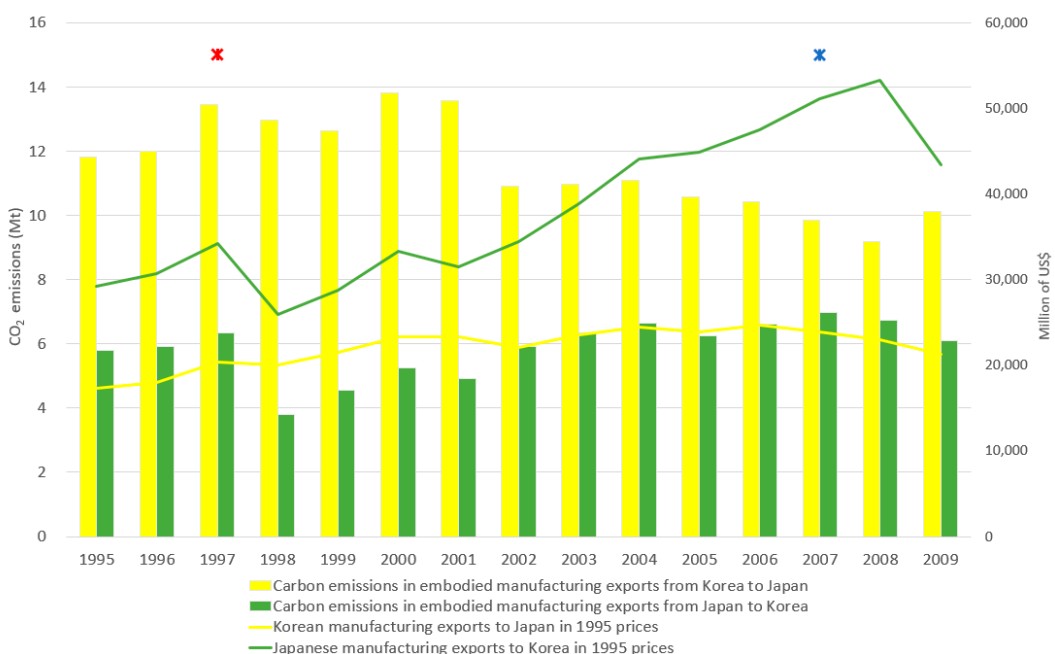

**Figure 3.** Exports and embodied $CO_2$ emissions in manufacturing trade between Korea and Japan. Note: A red marker indicates the Asian financial crisis, and a blue marker indicates the global financial crisis.

Overall, in embodied carbon trade, Japan and Korea are net importers against China, and Japan is a net importer against Korea. The relationship between exports and the carbon emissions embodied in exports shows a similar trend, but there is a difference in growth and movement, which move in opposite directions and make more drastic movements for certain periods. This trend indicates the complexity of the embodiment flow relationship between international trade and its effects on embodied $CO_2$ emissions, and embodied carbon emissions are jointly determined by various aspects of trade.

Figure 4 shows the net export of embodied emissions of the three countries. In particular, during 2000–2005, there were clear increases in net exports in trade between China and Japan and between China and Korea, and a decrease between Korea and Japan. Before the financial crisis, the carbon emissions embodied in China's exports increased drastically, while those of Japan and Korea increased only slightly, resulting in a significant increase in the imbalance. Based on the above results, it is considered that the possibility of carbon leakage is high between 2000 and 2005. Therefore, we used the SDA to identify various driving factors of such changes at the national and sectoral levels.

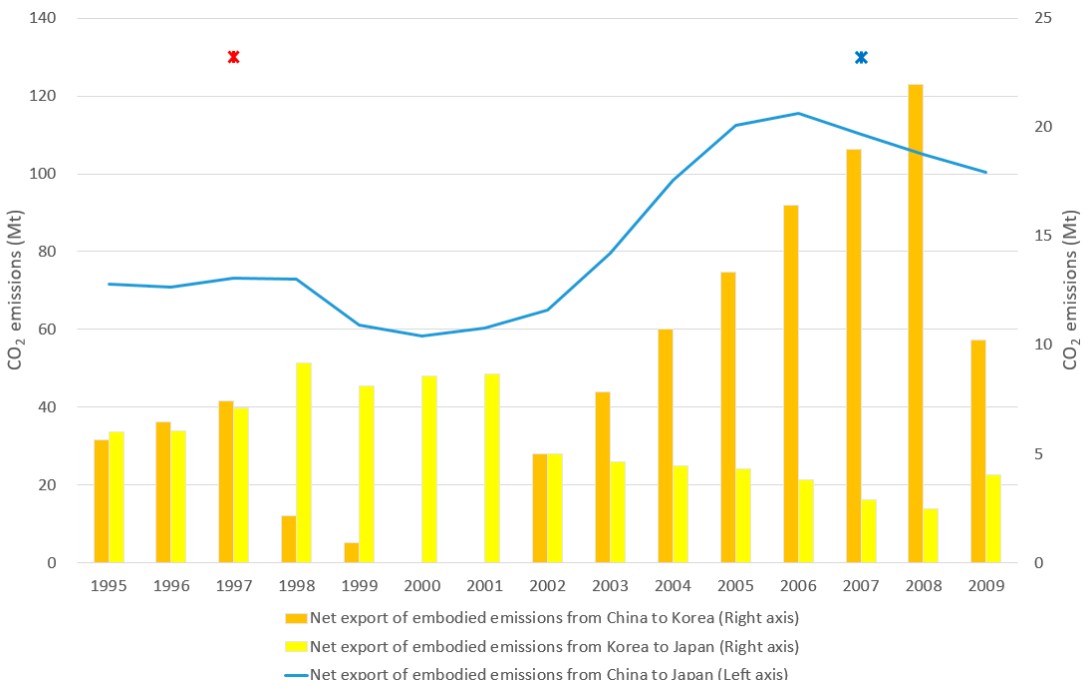

**Figure 4.** Net export of embodied $CO_2$ emissions in manufacturing trade among three countries. Note: A red marker indicates the Asian financial crisis, and a blue marker indicates the global financial crisis.

*4.2. Aggregate SDA Results*

The carbon emissions embodied in exports estimated in this study show that imbalances increased significantly among these countries in the period 2000–2005. Therefore, we divided the study period and focused on the 2000–2005 period, which showed rapidly increasing or decreasing trends in the estimated embodied emissions in bilateral trade, as shown in the previous section. The driving factors of $CO_2$ emissions embodied in manufacturing trade are quantified using the SDA technique. The main results of driving factors for the embodied emissions in the bilateral trade at the national level are as follows. Table 1 shows the contribution ratio for the changes in carbon emissions embodied in exports among countries over the 2000–2005 period. The columns show the percentage of changes in carbon emissions embodied in exports due to each of the 14 components, and the last column shows the total changes over the period.

The most striking feature of these results is the largest increase in embodied emissions in exports from China to Japan. During the period 2000–2005, the carbon emissions embodied in China's exports to Japan increased by approximately 61 Mt. Looking at driving factors, "the share of demand abroad of final product" ($P^{(c)}$) had the largest impacts on the corresponding increase in carbon emissions embodied in China's exports. In China's exports to Korea, "composition of intermediate goods and services at home" ($T^{(c)}$) and "the share of demand abroad of final product" ($P^{(c)}$) had the largest impacts on the increase in carbon emissions embodied in China's exports. On the other hand, imports from both Japan and Korea increased, albeit slightly, at 7 Mt and 19 Mt respectively, and "total final demand abroad" ($G^{(-c)}$) was the most influential factor in the increase. Therefore, Japan's and Korea's increases

in China's final product consumption led to an increase in China's carbon exports. This reflects that China increasingly participated in the production and exports of carbon-intensive goods and services.

**Table 1.** The contribution ratio of driving factors in embodied carbon emissions in exports during 2000–2005 (%).

| Driving Factors | China to Japan | Japan to China | China to Korea | Korea to China | Korea to Japan | Japan to Korea |
|---|---|---|---|---|---|---|
| Emission factors[(c)] | 1% | −15% | 1% | −3% | −4% | −23% |
| Energy structure[(c)] | 7% | −5% | 8% | −7% | −3% | −7% |
| Energy intensity[(c)] | −73% | −27% | −83% | −67% | −57% | −28% |
| Intermediate trade[(c)] | 22% | 15% | 33% | −20% | 13% | 13% |
| Production technology[(c)] | 5% | −34% | 2% | 7% | 24% | −35% |
| Final products trade[(c)] | 65% | −15% | 31% | 2% | 63% | 21% |
| Final demand[(c)] | 0% | 0% | 1% | 0% | 0% | −1% |
| Emission factors[(−c)] | 0% | 0% | 0% | 0% | 0% | 0% |
| Energy structure[(−c)] | 0% | 0% | 0% | 0% | 0% | 0% |
| Energy intensity[(−c)] | 0% | −2% | 0% | −2% | 0% | −1% |
| Intermediate trade[(−c)] | −2% | −2% | −7% | −1% | −3% | −4% |
| Production technology[(−c)] | −10% | 3% | 4% | 3% | −7% | 10% |
| Final products trade[(−c)] | −5% | 4% | −10% | 1% | 0% | −1% |
| Final demand[(−c)] | −10% | 78% | 20% | 86% | −26% | 55% |
| Total change | 61,064 | 6886 | 19,288 | 4450 | −3289 | 1019 |

Notes: Positive and negative signs represent the direction of embodied emission changes in exports; "emission factors" represents the $CO_2$ emissions per unit heat of each energy consumption, "energy structure" represents the share of each energy consumption, "energy intensity" represents the energy consumption per unit output, "intermediate trade" represents the composition of intermediate inputs, "production technology" represents the intermediate input per unit output, "final products trade" represents the share of final products in export market, and "final demand" represents the demand for the final product; the superscript (c) and superscript (−c) indicate the factors at home and abroad, respectively.

On the contrary, Japan's and Korea's increases in carbon exports were most affected by China's growth in final demand. This reflects the increase in export volume due to the increased demand from China's economic growth during 2000–2005. Meanwhile, in the trade between Korea and Japan, Korea's reduction in carbon exports was remarkable. During 2000–2005, "energy intensity" ($E^{(c)}$) had the largest negative impact, and embodied carbon in exports decreased at 3 Mt. In Japan's exports to Korea, embodied emissions slightly increased at 1 Mt. "Total final demand abroad" ($G^{(−c)}$) was the most influential factor in the corresponding increase. As a result, the imbalance between the two countries decreased slightly. We found similar results for the entire study period as in the case above. This shows that the increase in China's final demand has the largest impact on Japan's and Korea's carbon emissions embodied in exports to China. Overall, the results hint at the carbonization of the industry due to China's industrialization and strong economic growth with increased exports in the early 2000s.

*4.3. SDA Results at the Sectoral Level*

In this section, the sectoral SDA results of embodied emissions in manufacturing trade among the three countries are presented. As identified in the previous section, the changes in China's and Korea's carbon exports were remarkable during 2000–2005, the carbon emissions embodied in China's and Korea's exports were analyzed in this period. For meaningful analysis, only the top five manufacturing sectors in 2005 are displayed in Figures 5–7, and the other sectors with the remaining total amount of $CO_2$ emissions in exports are grouped into the category, "other sectors". In Figures 5–7, sectors are listed in descending order from the bottom, with the category "other sectors", which is always located at the top of the graph to be distinguishable.

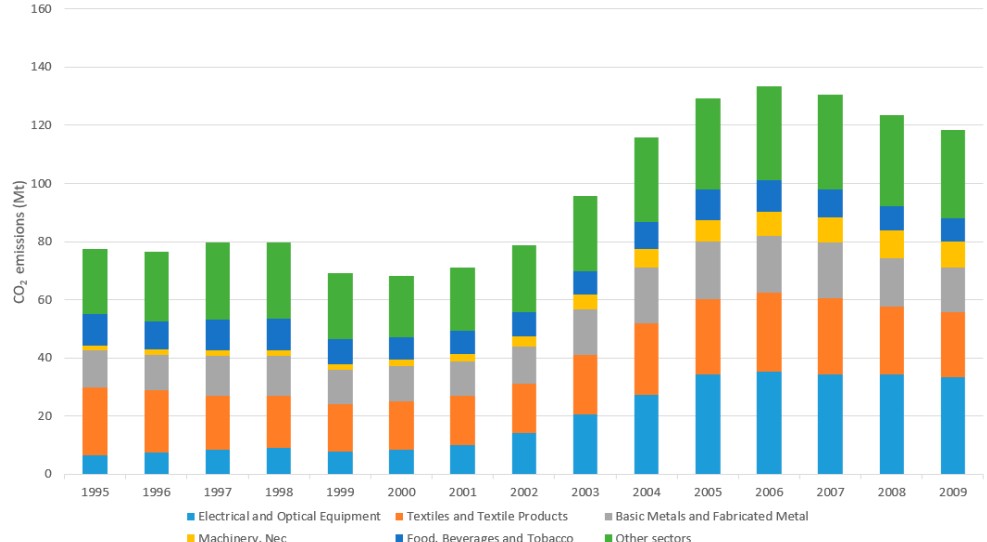

**Figure 5.** The share of sectors in embodied CO$_2$ emissions manufacturing exports from China to Japan.

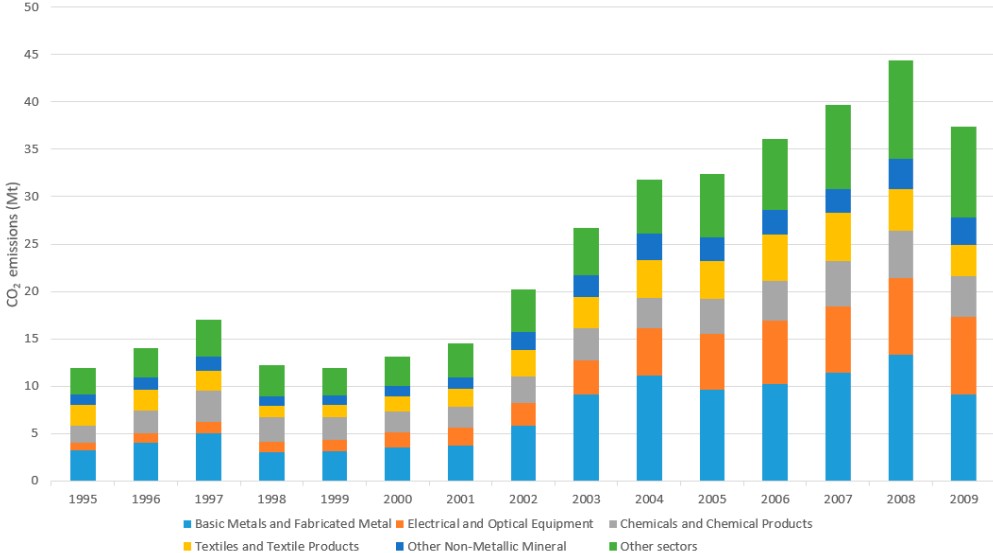

**Figure 6.** The share of sectors in embodied CO$_2$ emissions manufacturing exports from China to Korea.

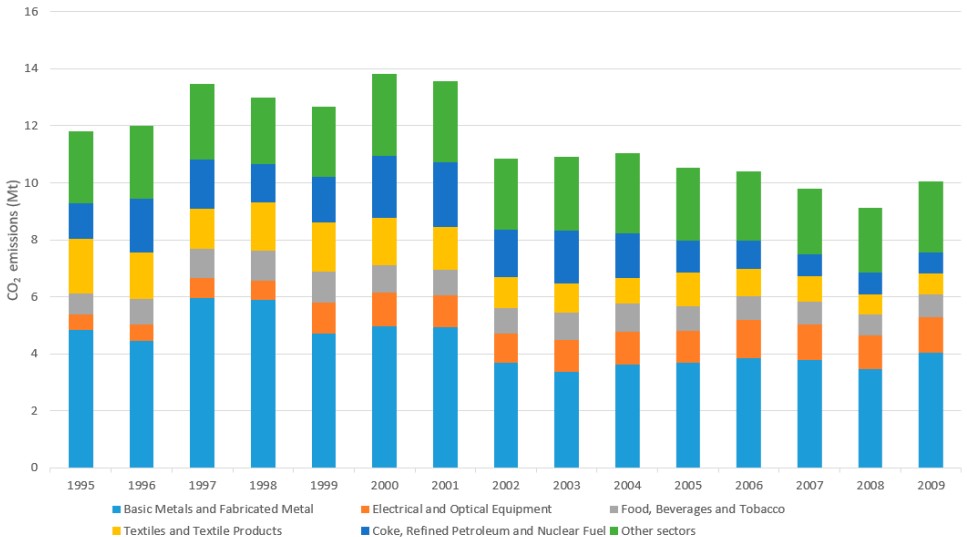

**Figure 7.** The share of sectors in embodied CO$_2$ emissions manufacturing exports from Korea to Japan.

### 4.3.1. China to Japan

Figure 5 shows the amount of each industry in the embodied carbon in exports from China to Japan during 1995–2009 and Table 2 represents the sectoral SDA results of the carbon emissions embodied in exports from China to Japan during 2000–2005. The columns show the top five sectors and the rows show the driving factors of emission changes.

**Table 2.** The sectoral driving factors on embodied carbon emissions in exports from China to Japan during 2000–2005 (kt).

|  | Food, Beverages, and Tobacco | Textiles and Textile Products | Basic Metals and Fabricated Metal | Machinery, Nec | Electrical and Optical Equipment |
|---|---|---|---|---|---|
| Emission factors[(c)] | 32 | 141 | −10 | 64 | 286 |
| Energy structure[(c)] | 185 | 1318 | 1048 | 370 | 1319 |
| Energy intensity[(c)] | −1419 | −2786 | −278 | −920 | −3043 |
| Intermediate trade[(c)] | 265 | 838 | 7299 | 520 | 552 |
| Production technology[(c)] | 325 | 2576 | −202 | 299 | −109 |
| Final products trade[(c)] | 3892 | 13,285 | 972 | 3825 | 23,242 |
| Final demand[(c)] | 1 | 6 | 139 | 9 | 12 |
| Emission factors[(−c)] | 0 | 0 | −1 | −2 | −9 |
| Energy structure[(−c)] | 0 | −1 | 0 | 0 | −3 |
| Energy intensity[(−c)] | −1 | −1 | 0 | −2 | −24 |
| Intermediate trade[(−c)] | −3 | −14 | −166 | 5 | 23 |
| Production technology[(−c)] | −13 | −35 | −961 | −52 | −97 |
| Final products trade[(−c)] | −7 | −160 | −313 | −18 | −30 |
| Final demand[(−c)] | −441 | −6062 | 153 | 1155 | 3883 |
| Total changes | 2815 | 9105 | 7679 | 5252 | 26,003 |

Notes: The superscript (c) and superscript (−c) indicate the factors at home and abroad, respectively.

In Chinese exports to Japan, regarding the change in carbon emissions embodied in exports at the sectoral level, the impact of specific industries was significant. "Electrical and Optical Equipment", "Textiles and Textile Products", and "Basic Metals and Fabricated Metal" account for a high share of the carbon emissions embodied in exports. "Electrical and Optical Equipment" showed the largest increase in carbon exports, and the increase was very noticeable between 2000 and 2005. It accounted for more than 42% (26 Mt of $CO_2$) of total growth (61 Mt of $CO_2$) during 2000–2005.

From Table 2, this increase was mostly influenced by "The share of demand abroad of final product" ($P^{(c)}$). In the rest of the industries, the influence of "the share of demand abroad of final product" ($P^{(c)}$) was also large, but in the "Basic Metals and Fabricated Metal" industry, "composition of intermediate goods and services at home" ($T^{(c)}$) was the most significant. "Energy intensity" ($E^{(c)}$) had the largest negative impact on carbon emissions in all industries and helped offset the increase in embodied carbon emissions. Overall, Japan's increase in consumption of China's final products in "Electrical and Optical Equipment" and "Textiles and Textile Products" led to an increase in China's carbon exports.

### 4.3.2. China to Korea

In Chinese exports to Korea, "Basic Metals and Fabricated Metal", "Electrical and Optical Equipment", and "Textiles and Textile Products" account for a high proportion of the carbon emissions embodied in exports in Figure 6. The largest increase was seen in "Basic Metals and Fabricated Metal" at 6 Mt and the increase was very noticeable between 2000 and 2005. It accounted for more than 31% (6 Mt of $CO_2$) of total growth (19 Mt of $CO_2$) during 2000–2005.

From Table 3, the sectoral SDA results are presented, and the driving factors affecting the embodied carbon emissions of each industry were very different. "Basic Metals and Fabricated Metal" was mostly

influenced by "composition of intermediate goods and services at home" ($T^{(c)}$) at 4 Mt. In the case of "Electrical and Optical Equipment", which had the second-largest influence at 4.2 Mt, "the share of demand abroad of final product" ($P^{(c)}$) was the most significant at 3.8 Mt. In the "Textiles and Textile Products", "total final demand abroad" ($G^{(-c)}$) was the most influential factor in the increase. On the other hand, "energy intensity" ($E^{(c)}$) had the largest negative impact on carbon emissions in all industries except for "Other Non-Metallic Mineral" and helped offset some increase.

**Table 3.** The sectoral driving factors on embodied carbon emissions in exports from China to Korea during 2000–2005 (kt).

| | Textiles and Textile Products | Chemicals and Chemical Products | Other Non-Metallic Mineral | Basic Metals and Fabricated Metal | Electrical and Optical Equipment |
|---|---|---|---|---|---|
| Emission factors$^{(c)}$ | 54 | 7 | 8 | 51 | 51 |
| Energy structure$^{(c)}$ | 233 | 293 | 31 | 564 | 320 |
| Energy intensity$^{(c)}$ | −429 | −879 | 71 | −10 | −728 |
| Intermediate trade$^{(c)}$ | 150 | 1195 | 927 | 3996 | 385 |
| Production technology$^{(c)}$ | 402 | −6 | −9 | −164 | −11 |
| Final products trade$^{(c)}$ | 293 | 247 | −34 | 94 | 3872 |
| Final demand$^{(c)}$ | 3 | 37 | 32 | 193 | 19 |
| Emission factors$^{(-c)}$ | −1 | 0 | 0 | 0 | 0 |
| Energy structure$^{(-c)}$ | −1 | 0 | 0 | 0 | −1 |
| Energy intensity$^{(-c)}$ | −6 | 0 | 0 | 0 | −9 |
| Intermediate trade$^{(-c)}$ | 27 | −86 | −26 | −224 | 33 |
| Production technology$^{(-c)}$ | 66 | 235 | 91 | 437 | 42 |
| Final products trade$^{(-c)}$ | −9 | −42 | −31 | −242 | −16 |
| Final demand$^{(-c)}$ | 1630 | 491 | 383 | 1397 | 276 |
| Total changes | 2411 | 1493 | 1442 | 6094 | 4233 |

Notes: The superscript (c) and superscript (−c) indicate the factors at home and abroad, respectively.

Meanwhile, unlike from China to Japan, China's exports to Korea have more carbon emissions embodied in exports of intermediate goods than in exports of final goods. The increase in exports of intermediate goods led to an increase in carbon emissions embodied in exports. However, the increase in carbon emissions embodied in exports of intermediate goods and services is relatively smaller than the increase in exports of intermediate goods and services. These results show that China is considered "the World's Factory", which mainly imports intermediate goods such as parts and raw materials and exports carbon-intensive final goods.

### 4.3.3. Korea to Japan

In Korean exports to Japan in Figure 7, "Basic Metals and Fabricated Metal" and "Coke, Refined Petroleum, and Nuclear Fuel" account for a high proportion of the carbon emissions embodied in exports and the reduction in these industries was the largest at 1.2 and 1 Mt, respectively, during 2000–2005. They accounted for more than 70% (2.2 Mt of $CO_2$) of total reduction (3.2 Mt of $CO_2$) during 2000–2005.

From the sectoral SDA results in Table 4, "Basic Metals and Fabricated Metal" was mostly influenced by "energy intensity" ($E^{(c)}$) at 1.2 Mt and "Coke, Refined Petroleum, and Nuclear Fuel" was mostly influenced by "composition of intermediate goods and services at home" ($T^{(c)}$) and "energy intensity" ($E^{(c)}$) at 0.46 and 0.43 Mt, respectively. This shows that high portion industries in Korea's embodied emissions in exports have decreased carbon emissions.

**Table 4.** The sectoral driving factors on embodied carbon emissions in exports from Korea to Japan during 2000–2005 (kt).

| | Textiles and Textile Products | Leather, Leather and Footwear | Coke, Refined Petroleum and Nuclear Fuel | Other Non-Metallic Mineral | Basic Metals and Fabricated Metal |
|---|---|---|---|---|---|
| Emission factors[(c)] | −44 | −4 | −1 | −5 | −69 |
| Energy structure[(c)] | −4 | 1 | 89 | −3 | −126 |
| Energy intensity[(c)] | −260 | −13 | −435 | −110 | −1236 |
| Intermediate trade[(c)] | −48 | −2 | −463 | −8 | 505 |
| Production technology[(c)] | 50 | 3 | 20 | 13 | 64 |
| Final products trade[(c)] | 726 | −83 | −221 | 0 | −4 |
| Final demand[(c)] | 0 | 0 | 0 | 0 | 5 |
| Emission factors[(−c)] | 0 | 0 | 0 | 0 | −1 |
| Energy structure[(−c)] | 0 | 0 | 0 | 0 | 0 |
| Energy intensity[(−c)] | −1 | 0 | 1 | 0 | 0 |
| Intermediate trade[(−c)] | −1 | 0 | −25 | −2 | −83 |
| Production technology[(−c)] | −1 | 0 | −55 | −21 | −145 |
| Final products trade[(−c)] | 0 | 0 | 1 | 0 | 1 |
| Final demand[(−c)] | −886 | −65 | 44 | −21 | −195 |
| Total changes | −470 | −164 | −1046 | −156 | −1283 |

Notes: The superscript (c) and superscript (−c) indicate the factors at home and abroad, respectively.

Unlike from China to Japan and Korea, Korea's exports to Japan show decreased embodied carbon emissions in exports, while Korea's exports to Japan are stable. There was also a reduction in the final demand for Korea's "Textiles and Textile Products", "Basic Metals and Fabricated Metal", and "Leather and Footwear", but increases in energy efficiency in all manufacturing industries led to a decrease in carbon emissions embodied in exports. Among the manufacturing industries, the increase in the energy efficiency of "Basic Metals and Fabricated Metal" was the most prominent, and it was found that Korea is steadily increasing the energy efficiency of the manufacturing industry during the analysis period.

## 5. Discussion and Conclusions

We estimated the embodied carbon emissions in manufacturing trade among China, Japan, and Korea during 1995–2009 using an environmental multiregional input–output model and analyzed the driving factors of changes in embodied carbon emissions. The main conclusions are as follows:

China was a net exporter of embodied carbon emission, despite a substantial manufacturing trade deficit with Japan and Korea. China's exports were much more carbon-intensive manufacturing products than its imports from Japan and Korea, and there was a significant imbalance of embodied emission in its trade with Japan and Korea. The largest imbalance was observed between China—a developing, Non-Annex I country—and Japan—a developed, Annex I country—and this imbalance continues to increase during 2000–2005. Regarding the change in carbon emissions embodied in exports, the impact of specific industries was significant. Japan's increase in consumption of China's final products in "Electrical and Optical Equipment" and "Textiles and Textile Products" led to an increase in China's carbon exports. Zhao, et al. [23] and Wu, et al. [24], who calculated carbon emissions from all industries embodied in the trade from China to Japan through domestic technology assumption, also shows similar results, with rapid increases in in "Electrical and Optical Equipment" and "Textiles and Textile Products".

"Electrical and Optical Equipment" showed the largest increase in carbon exports, and this increase was mostly influenced by "the share of demand abroad of final product" ($P^{(c)}$). "Energy intensity" ($E^{(c)}$) had the largest negative impact on carbon emissions in all industries and helped offset the increase in embodied carbon emissions.

Between China and Korea, the main driving factors for the increased embodied emissions in China's exports, which caused an imbalance, are the trade structure of intermediate and final products: "composition of intermediate goods and services at home" ($T^{(c)}$) and "the share of demand abroad

of final product" (P$^{(c)}$) in a few sectors, such as "Basic Metals and Fabricated Metal", "Electrical and Optical Equipment", and "Textiles and Textile products". Regarding Korea's and Japan's exports, their increased carbon emissions embodied in exports to China were mainly affected by China's total demand. These results signify that a considerable share of production and carbon emissions has shifted to China, resulting in a significant increase in exports and in embodied emissions; moreover, the increase in consumption of final products from China's economic growth seems to have affected the increase in emissions from Korea and Japan.

As a result, before the first commitment period of the Kyoto Protocol, a significant amount of carbon leakage had been confirmed from Japan and Korea to China. These results reflect China's industrialization, carbonization, and rapid economic growth since 2000 when China joined the WTO. Especially, the increase in "Electrical and Optical Equipment" and "Basic Metals and Fabricated Metal" was remarkable. These indicate China's increasing production of carbon-intensive export goods and services in the "Electrical and Optical Equipment" and "Basic Metals and Fabricated Metal" sectors and its unfavorable position in the division of labor compared with Japan and Korea in an environmental aspect.

From the above results, considerable carbon leakage existed in China and the current regime is not enough to respond to climate change. In particular, in the new climate system of the Paris Agreement, the possibility of carbon leakage is more likely due to the asymmetric greenhouse gas reduction policy among the three countries that are in competition among industries. The introduction of the carbon border tax through Europe's "Green Deal" is part of a policy to prevent this carbon leakage and protect the industrial competitiveness of the country. To implement efficient worldwide GHG reduction, it requires cooperation among countries considering the embodied emissions in international trade and externality of environmental problems.

To prevent carbon leakage and minimize the impact of overseas consumption on domestic carbon emissions, national responsibility should be changed and consider embodied emissions in both trade (exports and imports). In the results of this study, carbon leakage in certain industries was prominent such as "Electrical and Optical Equipment", "Basic Metals and Fabricated Metal", and "Textiles and Textile Products". Therefore, it is necessary to consider policies to prevent carbon leakage in these industries; furthermore, it is necessary to evaluate the possibility of carbon leakage in the new climate system by considering industrial characteristics such as carbon intensity, energy intensity, trade intensity, and replaceability, including maturity by industry.

From the results, there are also references to reduce greenhouse gas emissions domestically. Considering the economic size, Japan has relatively low energy consumption and carbon emissions compared to China. In particular, the very low embodied emissions in export—as compared with the increase in export volume—seem to reflect the relatively high proportion of natural gas use and high energy-use efficiency. Japan continues to improve its energy efficiency and reduce the use of fossil fuels. Thus, the total $CO_2$ emissions remained stable. Korea seems to have reduced carbon emissions embodied in exports through a steady improvement in energy efficiency. According to our results, the adjustment in energy efficiency contributed a 51% and 58% decrease in carbon emissions embodied in Korea's exports to Japan and China during 1995–2009.

On the other hand, due to industrialization, rapid economic growth, and high energy demand, China has established a coal-based energy supply structure. According to our results, the adjustment in China's energy consumption structure has contributed a 9% and 8% increase in carbon emissions embodied in exports to Japan and to Korea during the study period. Japan's and Korea's experiences of energy conversion, decarbonization, and optimization of the energy consumption structure can provide policy implications for China's low-carbon economy. One possible way to mitigate embodied carbon emissions is to cooperate on environmentally friendly production technology in the carbon leakage industry. Japan and Korea are relatively energy-efficient compared with China. Thus, they can contribute to China's emission reduction through technical cooperation as consumers. In this way, carbon imbalances in trade can be reduced, essentially minimizing the inequity among countries in

carbon emission responsibility. Therefore, it is necessary to actively utilize market mechanisms such as the Sustainable Development Mechanism (SDM) in the Paris Agreement, which can be considered a method of domestic emission reduction.

Another option is ensuring balance in carbon prices, which can be achieved by linking the carbon market among the three countries operating the Emissions Trading System. By imposing balanced reduction costs, not only can carbon leakage due to unbalanced regulations be prevented, but companies can also be enabled to cost-effectively reduce emissions through the active carbon market. The campaign "RE100 Initiative", committing to use 100% renewable energy, could be a good alternative for the three countries. Many companies are currently participating in the campaign. Additionally, partners in the supply chain are required to use renewable energy, indirectly internalizing external costs, which could ultimately reduce total carbon leakage.

As we have presented in the previous sections, this study could contribute to the literature in terms of the decomposition model and the case study of Korea, Japan, and China. With the increasing interest in the embodied carbon emissions, future research would pay more attention to the bilateral trades of major carbon-emitting countries and multilateral trades. The analysis combining multiple databases could be another further research direction since it allows us to investigate recent changes. Our conclusion would be robust even if we could expand the study period into the mid-2010s. However, reshoring and protectionism are getting significant recently, and these could cause different results. This will also be another future research topic when the late-2010s data is available.

**Author Contributions:** Conceptualization, J.K. and Y.Y.; Methodology, Y.Y., Y.-K.K. and J.K.; Validation, J.K. and Y.-K.K.; Formal Analysis, Y.Y. and J.K.; Investigation, Y.Y.; Data Curation, Y.Y.; Writing—Original Draft Preparation, Y.Y.; Writing—Review and Editing, J.K. and Y.-K.K.; Supervision, J.K. and Y.-K.K.; Funding Acquisition, J.K. All authors have read and agreed to the published version of the manuscript.

**Funding:** This work was supported by the Human Resources Development program (No. 20194010201860) of the Korea Institute of Energy Technology Evaluation and Planning (KETEP), grant funded by the Korean Government Ministry of Trade, Industry and Energy.

**Conflicts of Interest:** The authors declare that there are no conflict of interest. The funders had no role in the design of the study, in the data collection or analyses, in the writing of the manuscript, or in the resulting and publishing the results.

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
