# Peer review of "Embodied CO2 Emission Changes in Manufacturing Trade: Structural Decomposition Analysis of China, Japan, and Korea"

_atmosphere, doi:10.3390/atmos11060597_

Round 1

Reviewer 1 Report

In the manuscript ID: atmosphere-799500, entitled: “Embodied CO2 emission changes in manufacturing trade: Structural decomposition analysis of China, Japan, and Korea”, the authors determined the carbon emissions embodied in manufacturing trade among (i) China, (ii) Japan and (iii) Korea. The authors used a Multi-Region Input-Output approach. The authors also used the SDA technique to analyze the driving factors of changes in embodied emissions (1995-2009).

The manuscript is of interest, well written and with a good background: for this reason, I suggest minor revisions before publication. Some specific comments are reported below:

  • Can the authors better explain why it was chosen to investigate only the period 1995-2009?
  • Figure 1-3: I suggest inserting a line in the graphs, related to some events reported in the text (the period affected by the Asian financial crisis and the period affected by the global financial crisis).
  • Figure 4: In my opinion the graph is not very clear and readable.
  • Figure 5-7: I suggest using another graph (histogram?). I also suggest making the caption clearer.
  • Discussions and Conclusions: I'd like to see more references and comparisons with them in the discussions.

Author Response

We thank the peer reviewers for helpful suggestions. We have taken almost all of them to heart. In terms of specific point-by-point responses, please see the "Response to reviewer.pdf" file.

Reviewer 2 Report

A very sophisticated presentation. I cannot se any need of changes.

Author Response

(The authors gave the same response as above.)

Reviewer 3 Report

The article “Embodied CO2 emission changes in manufacturing trade: Structural decomposition analysis of China, Japan, and Korea” provides a very interesting analysis on the driving factors of embodied carbon emission changes in manufacturing trades through structural decomposition analysis, showing how China is a net exporter of embodied carbon emissions to Japan and Korea, despite a substantial trade deficit.

The paper is well organized and developed. Prior to be published I suggest expanding a little bit the discussion section, including a comparison between authors’ results and previous studies.

Author Response

(The authors gave the same response as above.)

Reviewer 4 Report

  1. The article is based on 1995-2009 statistics. These are old data that are not very relevant today. Therefore, the article must be based on the latest data or at least statistics from the last decade. Or, in principle, those data must be directly correlated with today's issues, and with a justification of how they affected the current situation.
  2. The annotation is not representative.
  3. In the introduction reference is made to Annex I, but the end of the article is the Appendix A. The same in 15 p. line 470. So how is it really?
  4. View journal citation requirements.
  5. In 3 p. line 99 it is used the abbreviation SDA. First mentioned, it must be explained in full. The same is 9 p. line 340 (MG).
  6. At the beginning of Chapter 2, i. before subsections 2.1 it is recommended to write an introduction (explanation) of what methods and why they are used, what is the logical sequence of the description of the research methodology. There is also a lack of transition between subsection.
  7. In 4 p. line 151 it is written "........ and imports-equation, are eliminated." It should be explained why it is eliminated.
  8. The description of the methodology does not refer to the sources from which the formula is taken.
  9. In 5 p. line 202 it is written "....theses 14 factors." What they are?
  10. Formulas 8 and 9 are given, but in their explanations the variables are given differently, arguing that “.... we decomposed ....” Perhaps the final formulas should be given, what changes after decomposed?
  11. In 5 p. line 223, reference is made to source 14. Doubts or a good link because the authors and content are different.
  12. It is recommended to adjust the title of the 3 chapters.
  13. Moreover, this section is not informative, simply provides general information and does not correspond to the content of the existing title.
  14. In 6 p. line 241 it is written "....price tables" and it it given reference to 17 literature source. Incorrect information provided. 
  15. Move the figures and tables closer to the links in the text.
  16. References to sources are missing below the figures.
  17. In 9 p. lines 332-343 the facts are stated, but there is no analysis and insight from the authors of the article as to what is out of it. E.g. Line 332 states ".... largest increase ....". Question - why did it happen? and etc.
  18. In section 4 the facts are stated, but there is no analysis and insight from the authors of the article as to what is out of it, what to do and so on.
  19. If statistics from 1995 to 2009 are used, it is not appropriate to state that ".... China's exports are ...." ( 15 p. line 467) because this has all been the case in the past.
  20. A link to the RE100 Initiative appears at the end of the article. It is necessary to explain briefly what this initiative is and what its essence is, what it is specifically relevant to the issue at hand, and not to state that it is a good thing.
  21. There is no point in adding Appendix A, just at the beginning of the article it is possible to provide this information not in the table but in the text, come up with an abbreviation for it and continue to use it in the text.

Author Response

(The authors gave the same response as above.)

Round 2

Reviewer 4 Report

  1. It is recommended to write at the end of the abstract where and or what research can be further developed on the topic
  2. In my opinion, this sentence should be adjusted "In particular, under the current Kyoto Protocol, territorial-based national greenhouse gas (GHG) inventories assign responsibility for GHG emissions to the producing countries, and Annex I countries have been provided incentives to reduce their emissions through international trade with Non-Annex I countries [1].", whereas reference is made to Annex I and Non-Annex I and reference is made to the literature source (I do not think that the reader will look for the source in question to see Annex I and Non-Annex I)
  3. At the end of the discussion and conclusions section, it is also recommended to place emphasis on further research directions on the issue under consideration.

Author Response

We truly appreciate you taking the time to read our manuscript one more time and giving helpful suggestions.

1. It is recommended to write at the end of the abstract where and or what research can be further developed on the topic

(Answer) Good point. We added one sentence for the future research direction in the abstract.

2. In my opinion, this sentence should be adjusted "In particular, under the current Kyoto Protocol, territorial-based national greenhouse gas (GHG) inventories assign responsibility for GHG emissions to the producing countries, and Annex I countries have been provided incentives to reduce their emissions through international trade with Non-Annex I countries [1].", whereas reference is made to Annex I and Non-Annex I and reference is made to the literature source (I do not think that the reader will look for the source in question to see Annex I and Non-Annex I)

(Answer) Agreed. Following your suggestion, we changed this sentence into "In particular, under the current Kyoto Protocol, territorial-based national greenhouse gas (GHG) inventories assign responsibility for GHG emissions to the producing countries, and countries with reduction obligations have been provided incentives to reduce their emissions through international trade with other countries with less stringent emission constraints [1]."

3. At the end of the discussion and conclusions section, it is also recommended to place emphasis on further research directions on the issue under consideration.

(Answer) Good suggestion. We added three sentences for further research directions at the end of the discussion and conclusions section.